# How Quotation Types Shape Classic Novel Reading in Chinese: A Comparison Between Human Eye-Movements and Large Language Models

**DOI:** 10.3390/bs15121650

**Published:** 2025-11-30

**Authors:** Lijuan Chen, Wenjia Zuo, Xiaodong Xu

**Affiliations:** 1School of Foreign Studies, Nanjing University of Posts and Telecommunications, Nanjing 210023, China; 20210009@njupt.edu.cn; 2School of Foreign Languages and Cultures, Nanjing Normal University, Nanjing 210097, China

**Keywords:** quotation types, narrative perspective, direct speech, free indirect speech, eye-tracking, large language models, *surprisal*, *entropy*

## Abstract

Quotations play a central role in shaping narrative perspective, as they guide readers’ adoption and shifting of character and narrator viewpoints. While direct speech (DS) is often assumed to enhance vividness and emotional engagement, its cognitive demands relative to free direct speech (FDS) and free indirect speech (FIS) remain unclear, particularly in Chinese classical literature. Using eye-tracking, we investigated how Chinese readers process DS, FDS, and FIS in the Four Great Classical Novels, manipulating perspective congruency through *address terms* versus *proper names*. The results revealed two key findings. First, DS consistently incurred longer fixation times than FIS, demonstrating its higher processing cost. Second, congruency effects emerged robustly across all quotation types (including FIS) in later measures, suggesting that in the specific context of classical Chinese novels, FIS does not exhibit the dual-voice effect proposed in narrative theory for this particular manipulation. Complementary analyses with large language models (LLMs) further showed that DS yielded higher *surprisal* and *entropy* than both FDS and FIS, indicating greater contextual unpredictability. By integrating human eye-movement evidence with computational modeling, this study provides evidence about the cognitive processing of DS in Chinese classical texts and raises questions about the universality of dual-voice processing in FIS across different languages and text types.

## 1. Introduction

Quotations in narrative discourse provide a crucial interface between language and cognition because they explicitly encode how perspective is constructed and interpreted. Since Plato’s classical distinction between *diegesis* and *mimesis*, scholars have emphasized that quotation types—most prominently direct speech, free direct speech, and free indirect speech—differ not only in their formal and functional properties but also in how they position readers with respect to the narrator and the character ([31]; [39]). These differences underscore the central role of narrative perspective, defined as the vantage point from which events and mental states are represented. Whereas direct and free direct speech typically immerse readers in the character’s immediate perspective, free indirect speech creates a more complex blend of narrator and character voices, raising long-standing debates about whether readers adopt a single or dual perspective (or dual voice). From a cognitive standpoint, investigating how readers process these quotation types is essential for understanding how narrative discourse guides perspective-taking, shapes empathy, and modulates the allocation of cognitive resources during comprehension.

From the perspective of embodied cognition ([5], [6]), understanding quotations involves not merely processing abstract linguistic symbols but activating sensorimotor simulations that ground meaning in bodily experience. This theoretical framework suggests that when readers encounter direct speech, they may engage in more vivid perceptual and motor simulations of the speaking character’s voice and gestures, while indirect forms may rely more on abstract conceptual processing. Additionally, *Theory of Mind* (ToM)—the cognitive ability to understand and attribute mental states to others—plays a crucial role in narrative comprehension, particularly when processing different perspectives embedded in quotation types ([27], [28]; [37]). The integration of these theoretical perspectives suggests that quotation processing involves complex interactions between embodied simulation processes and mentalizing abilities.

To date, however, most research on quotation and reported speech has focused on differences in linguistic form, stylistic function, and rhetorical effect, often relying on non-experimental and interpretive approaches. Much less is known about the cognitive mechanisms underlying the processing of different quotation types, particularly in the context of Chinese classic novels, where quotations are stylistically diverse and play a pivotal role in shaping narrative perspective. Furthermore, the field has seen significant theoretical and methodological advances in recent years, including developments in computational narratology ([40]), advances in understanding perspective-taking mechanisms ([1]), and new insights into the neural substrates of communicative action understanding ([55]). Despite these advances, the application to Chinese narrative texts remains limited. To address this gap, the present study employs eye-tracking as an online measure of reading behavior, complemented by *surprisal* and *entropy* estimates derived from large language models. These measures capture contextual predictability and uncertainty, allowing us to examine whether reading-time differences reflect intrinsic processing demands or broader discourse expectations, and to compare how humans and computational models handle perspective in narrative comprehension.

### 1.1. Quotation and Narrative Perspective

Narrative perspective refers to the vantage point from which a story is told ([23]; [43]). Direct speech is conventionally anchored in the character’s perspective: first-person pronouns, present-tense verbs, and deictic expressions are interpreted relative to the character’s here-and-now. Free direct speech retains this anchoring but with greater formal economy, dispensing with explicit markers of narration. Both forms facilitate alignment with the character’s inner voice and are thought to enhance empathy and vividness ([59]; [63]). From an embodied cognition perspective, these forms may trigger what [6] ([6]) calls “perceptual symbols,” activating modal representations that simulate the sensory and motor aspects of the character’s speech act.

Indirect speech, in contrast, represents the character’s words or thoughts through the narrator’s perspective. It typically involves the use of third-person pronouns, tense backshifting, and adjusted deictic expressions, which reframe the original utterance within the narrator’s temporal and spatial coordinates. This makes indirect speech less immediate than direct speech, but it enables smoother integration into the surrounding narrative discourse and offers the narrator greater interpretive control ([3]; [13]; [52]). Moreover, indirect speech has been argued to reduce vividness and empathy compared to direct speech, while supporting narrative coherence and expository clarity ([20]; [26]). Free indirect speech (FIS), by contrast, occupies an intermediate position between narrator and character. It typically employs third-person pronouns and back-shifted tense, aligning with the narrator’s grammar, while preserving expressive markers (exclamatives, dialectal features, adverbials of time and place) that convey the flavor of direct speech ([3]; [31]; [45]). This hybrid form has motivated two competing accounts: *the single-voice theory*, which attributes FIS either to the narrator or to the character, and *the dual-voice theory*, which posits that readers experience both voices simultaneously ([9]; [17]; [46]). The dual-voice theory has been especially influential, suggesting that FIS places unique demands on readers by requiring the integration of two perspectives. Recent theoretical work has further distinguished between FIS and related phenomena such as ‘protagonist projection’ ([1]; [54]), where perspective shift occurs without presumed speech acts, adding nuance to our understanding of perspective-taking mechanisms in narrative. This distinction is particularly relevant for cross-linguistic research, as the linguistic resources available for marking perspective vary considerably across languages. In Chinese, for instance, the absence of grammatical tense and the prevalence of aspect markers may facilitate viewpoint shifts without the typical morphosyntactic cues associated with FIS in Indo-European languages.

### 1.2. Empirical Studies of Quotation and Perspective

#### 1.2.1. Direct and Indirect Speech

Empirical research consistently demonstrates that direct speech (DS) and indirect speech (IS)—despite often conveying the same propositional content—elicit distinct cognitive and affective responses. DS is associated with richer perceptual and phonological simulation as well as stronger emotional engagement, whereas IS promotes more abstract, resource-efficient semantic encoding ([12]; [30]; [63]). These patterns have been observed across adult reading experiments, developmental studies, and clinical populations, pointing to robust cross-method effects that call for integrative mechanistic explanations.

Research highlights a fundamental divergence in simulation mechanisms between DS and IS. DS reliably activates multimodal perceptual systems, constructing vivid scene representations consistent with embodied cognition accounts ([63], [64]). This aligns with the grounded cognition framework ([7]), which proposes that cognition emerges from interactions between classic cognitive processes, sensorimotor systems, and environmental context. For instance, [63] ([63]) found that the emotional tone of DS modulated reading speed: sad contexts (e.g., funerals: …Slowly, he looked around and said: “*I’m grateful you’re all here. This is the end of the journey…*”) slowed reading, whereas tense contexts (e.g., competitions: …His mother encouraged him but he was all shaking and said: “*No! I can’t do it! This is the end of the journey because…*”) accelerated it. This effect was absent for IS (see also [53]). Similarly, phonological interference paradigms show that DS is more susceptible than IS to tongue-twister interference during silent reading, suggesting stronger activation of inner speech ([61]). Quotation choice is also shaped by psychological distance: DS is preferred for socially or temporally proximal events, while IS is favored when distance is greater ([33]). Neuroimaging studies support this account, showing that DS more strongly activates voice-selective auditory regions and increases theta phase-locking—both signatures of covert speech simulation—whereas IS involves weaker auditory engagement and is processed more abstractly ([2]; [62]; [65]).

At the same time, DS appears more cognitively demanding. Under non-interference conditions (without a memory task), producing DS is typically faster than IS, but when a secondary memory task is introduced, DS performance declines sharply, sometimes falling below IS ([34]). [30] ([30]) likewise found that DS increased error rates and response times in pronoun resolution tasks, particularly with third-person pronouns, suggesting heavier working memory demands. Probe recognition studies further indicate that DS does not necessarily enhance recall of textual content, and in some cases IS yields superior memory accuracy ([18]). Source memory studies also show that IS better preserves certain character attributes, such as gender ([19]). Together, these results suggest that while DS enhances perceptual vividness and emotional salience, it imposes higher cognitive costs, especially under resource-limited conditions. These findings suggest a tension between the embodied simulation processes activated by DS and the cognitive resources required to maintain these simulations, particularly when Theory of Mind processes are engaged to track speaker perspectives ([56]).

#### 1.2.2. Free Direct and Free Indirect Speech

Most empirical studies have focused on DS and IS, with relatively little attention to marginal forms such as free direct speech (FDS) and free indirect speech (FIS). FIS is particularly distinctive in blending narrator and character voices, producing blurred semantic–pragmatic boundaries and potentially unique processing mechanisms ([21]). While FIS has been theorized to involve a “dual voice effect” ([42]), empirical evidence remains sparse and mixed ([15]).

Existing studies on FIS have mainly used offline questionnaires. For example, [51] ([51]) found that about one-third of participants identified both narrator and character perspectives in FIS passages, comparable to those selecting the narrator alone. [9] ([9]) similarly reported nearly equal numbers of participants endorsing a dual-voice or character-only interpretation. [42] ([42]) found that unlike first-person narration and psycho-narration, FIS did not strongly encourage perspective-taking. Recent theoretical work suggests that what appears as FIS may sometimes be better characterized as protagonist projection or viewpoint shift ([1]; [58]), phenomena that give linguistic form to pre-verbal perceptual content without necessarily involving dual voicing. This distinction has important implications for understanding the cognitive processing of these forms.

#### 1.2.3. Chinese Narrative and Cultural Context

The study of quotation in Chinese presents unique opportunities and challenges. Chinese linguistic features—such as the absence of overt tense marking, frequent subject omission, and heavy reliance on contextual cues—create different conditions for perspective marking compared to Indo-European languages ([24]; [47], [48]). Moreover, the Four Great Classical Novels represent a distinctive narrative tradition with their own conventions for representing speech and thought. These texts, deeply embedded in Chinese cultural consciousness through education and adaptation, carry specific cultural expectations about character relationships and appropriate forms of address that may influence processing ([35]; [50]).

Recent work on Chinese narrative discourse has highlighted the role of cultural schemas and social hierarchies in shaping perspective-taking ([11]). The use of kinship terms versus proper names in addressing family members, for instance, carries strong pragmatic implications in Chinese that may not translate directly to other cultural contexts. This cultural specificity raises important questions about the universality of findings from Western language studies and highlights the need for culturally situated investigations of narrative processing.

The formal properties of FIS in Chinese classical literature warrant particular attention. Chinese classical novels, especially the Four Great Classical Novels examined in this study, employ distinctive narrative techniques that blur the boundaries between narrator and character perspectives. Unlike Western FIS, which relies on tense backshifting and pronominal reference to signal dual voicing, Chinese FIS often manifests through perceptual verbs (e.g., 见 ‘see’, 听闻 ‘hear’) and spatial-temporal deixis that orient the reader to a character’s experiential viewpoint while maintaining third-person narrative framing ([21]; [47]). These features suggest that what appears as FIS in Chinese may function more as a viewpoint-shifting device—in line with [1] ([1]) protagonist projection—than as genuine dual-voiced discourse. This linguistic and stylistic specificity raises important questions about whether the dual-voice effects reported for Western languages ([9]; [51]) extend to Chinese narrative contexts, making Chinese classical novels an ideal testing ground for theories of perspective-taking in narrative comprehension.

### 1.3. The Present Study

Building on these debates and findings, the present study examines how Chinese readers process direct speech (DS), free direct speech (FDS), and free indirect speech (FIS) during natural reading of classical Chinese novels. Using eye-tracking, we address three core questions: (1) Does the perspectival ambiguity of FIS, and its potential for maintaining two narrative perspectives, increase processing costs relative to DS and FDS in the context of Chinese classical literature? (2) How does perspective incongruency—mismatch between the narrator’s and the character’s perspective—affect comprehension speed across quotation types when operationalized through *address terms* versus *proper names*? (3) To what extent do large language models (LLMs) parallel humans in distinguishing quotation types and detecting perspective congruency in Chinese narrative texts? It is important to note that our operationalization of perspective congruency through *address terms* versus *proper names* represents one specific way to test perspective effects, with inherent limitations. This manipulation may not capture the full range of perspectival cues available in narrative texts, and the findings should be interpreted within this specific methodological context. Chinese, however, presents distinctive challenges: it lacks overt tense marking, often omits subjects, and relies heavily on pronouns as perspective cues (e.g., first-person pronouns for FDS, third-person for FIS). Cultural conventions also shape interpretation—for example, addressing one’s father by name is pragmatically infelicitous, making kinship terms such as *fuwang* (“father”) crucial markers of perspective congruency ([47]). Investigating quotation in Chinese therefore broadens the empirical base while testing the cross-linguistic generality of theoretical claims about quotation and perspective. 

Our materials were adapted from the *Four Great Classical Novels* and manipulated perspective congruency through *terms of address* versus *proper names*. The choice of classical novels as materials has both advantages and limitations. While these texts provide ecologically valid narrative materials with rich quotation patterns, they may differ from contemporary fiction in style and processing demands. Additionally, participants’ familiarity with these canonical texts may influence reading patterns in ways that would not occur with unfamiliar materials. We employed fine-grained eye-tracking measures (First Fixation Duration, Gaze Duration, Regression Path Duration, and Total Reading Time) to capture both early and late stages of processing. Based on prior work and embodied cognition theory, we predict an interaction between quotation type and congruency. Specifically, DS is expected to elicit longer reading times than FDS and FIS, reflecting greater perspective-taking demands and more intensive embodied simulation ([12]; [30]; [64]). Since the character’s perspective is regarded as the default in DS and FDS ([3]; [31]; [47]), the use of a reference form incongruent with this perspective (e.g., a proper name *Lijing* rather than *father*) should prolong reading times in these quotation types. Predictions for FIS are more complex: if FIS activates a genuine “dual voice”, then reference forms congruent with either the character’s or the narrator’s perspective should be processed similarly; however, if one perspective is privileged, incongruent reference forms should elicit longer reading times.

In addition to human data, we incorporate *surprisal* and *entropy* estimates from GPT-2 to compare computational predictions with human eye movements ([25]; [32]; [38]). While GPT-2 is not the most recent language model available, it was chosen for its open-source availability, computational tractability, and widespread use in psycholinguistic research, allowing for replicability and comparison with other studies. If both humans and GPT-2 exhibit similar sensitivity to quotation type and perspective congruency, this would suggest that LLMs encode discourse-level structures in ways that approximate human cognition. The FIS condition is particularly crucial: if GPT-2, like humans, represents dual voices, then significant differences should emerge between congruent and incongruent conditions; if not, no such difference is expected. Conversely, divergences between humans and the model would highlight aspects of perspective-taking and narrative comprehension that remain uniquely human. By integrating eye-tracking measures with computational estimates, this study moves beyond behavioral description to provide a theoretically grounded test of predictive processing in both human and artificial systems. This dual approach allows us to evaluate whether quotation types are processed merely as surface-level lexical patterns or whether they elicit deeper, coherence-driven expectations that distinguish human narrative understanding from model-based text generation.

## 2. Materials and Methods

### 2.1. Participants

An a priori power analysis was conducted using G*Power (version 3.1.9.7; [22]) to estimate the required sample size. Based on a repeated-measures ANOVA with a 2 × 3 within-subjects design (six measurements), assuming a medium effect size (*f* = 0.25), an alpha level of 0.05, and desired power of 0.95, the analysis indicated that a minimum of 28 participants was needed. We recruited 37 students (4 males; M = 24 years, SD = 2.0) from different universities. All reported normal or corrected-to-normal vision and no history of reading impairments. Participants provided written informed consent prior to the study and received monetary compensation upon completion. Each session lasted approximately 50 minutes. Data from four participants were excluded due to poor eye-tracking calibration (n = 3) or low comprehension accuracy (<80%; n = 1), resulting in a final sample of 33 participants.

### 2.2. Design and Materials

The experiment employed a 3 × 2 within-subjects design, crossing *quotation type* (DS: direct speech; FDS: free direct speech; FIS: free indirect speech) with *perspective congruency* (congruent vs. incongruent). Seventy experimental passages were adapted from the Four Great Classical Novels (*Journey to the West*, *A Dream of Red Mansions*, *Water Margin*, *and Romance of the Three Kingdoms*). These texts were chosen because they represent canonical works in Chinese literature with which educated Chinese readers have substantial familiarity, and they contain rich examples of different quotation types. However, we acknowledge that participants’ background knowledge and emotional engagement with specific characters may have influenced processing in ways not captured by our design ([16]). Each passage consisted of a context and a critical sentence (see Table 1). The context provided background information about the plot and character relationships. The critical sentence contained three clauses: the first described the mental state or action of a main character (e.g., Nezha, a young general in *Journey to the West*); the second reported the character’s speech, beginning with a critical pronoun—first person (“I”) in DS and FDS, or third person (“he/she”) in FIS; the third clause also reported speech but manipulated reference forms. In the congruent condition, the character used an appropriate *term of address* (e.g., *fuwang* “father”), whereas in the incongruent condition, the same referent was expressed with a *proper name* (e.g., *Lijing*, Nezha’s father).

Perspective congruency was defined with respect to the narrative viewpoint—that is, whether a quotation aligned with the perspective conventionally associated with a given speech type. Direct Speech (DS) and Free Direct Speech (FDS) are typically anchored in the character’s perspective, making the use of a proper name (e.g., *Nezha* calling his father “*Lijing*”) pragmatically odd and culturally inappropriate in Chinese (as confirmed by our acceptability tests; see Figure 1A). Free Indirect Speech (FIS), however, occupies a hybrid status, and it remains debated whether a proper name reflects the narrator’s or the character’s viewpoint. To maintain consistency and facilitate comparison across conditions, and following previous studies ([14]; [36]; [57]), we treated terms of address as indicators of the character’s perspective, even when preceded by a third-person pronoun (i.e., in the FIS condition). Accordingly, in our design, terms of address were coded as *congruent* and proper names as *incongruent*.

To select suitable stimuli, two pre-tests were conducted: a *text acceptability test* and a *narrative perspective test*. Based on the results, 48 passages were retained for the main experiment. Congruent items with mean acceptability ratings below 4 across the three quotation types were excluded, as were incongruent items with ratings above 5 in DS and FDS but not in FIS, since most incongruent free indirect items were rated nearly as highly as congruent ones. In addition, 12 filler passages were included. Like the experimental items, each filler consisted of a context and a three-clause critical sentence of comparable length. However, fillers were written as plain narration without quotation. To increase variability, half of the fillers were plausible within the plot, while the other half were deliberately implausible.

In the acceptability test, sixty participants were asked to read through the context and then judge the acceptability of the critical sentence using a 7-point rating scale (1 = least acceptable and 7 = most acceptable). The linear mixed-effects model on the selected 48 material sets revealed a significant interaction between quotation type and congruency (*χ*^2^ = 31.65, *p* < 0.001; See Figure 1A). Follow-up analyses showed that congruent passages were rated higher in acceptability than incongruent ones across all speech types—DS (*β* = 3.35, SE = 0.273, *t* = 12.25, *p* < 0.001), FDS (*β* = 2.99, *SE* = 0.273, *t* = 10.92, *p* < 0.001), and FIS (*β* = 1.95, SE = 0.273, *t* = 7.15, *p* < 0.001). Within the incongruent condition, FIS received higher ratings than both DS (*β* = −1.91, *SE* = 0.181, *t* = −5.05, *p* < 0.001) and FDS (*β* = −1.26, *SE* = 0.181, *t* = −6.93, *p* < 0.001), while DS and FDS did not differ significantly (*β* = 0.34, *SE* = 0.181, *t* = 1.89, *p* = 0.143). In the congruent condition, DS was rated higher than FDS (*β* = 0.71, *SE* = 0.181, *t* = 3.90, *p* < 0.001) and FIS (*β* = 0.48, *SE* = 0.181, *t* = 2.66, *p* < 0.05), whereas FDS and FIS did not differ (*β* = −0.22, *SE* = 0.181, *t* = −1.24, *p* = 0.431).

In the narrative perspective test, only materials from the congruent condition were used. Three lists were constructed, each containing 48 experimental items and 12 fillers, presented in a pseudorandomized order. Thirty-six participants completed one of the three lists. On each trial, participants read the context and indicated whose perspective was represented in the critical sentence. They were given three options: the narrator’s perspective, the character’s perspective, both the narrator’s and the character’s perspectives. Results are summarized in Table 2 and Figure 1B. The analysis revealed a significant interaction between quotation type and participants’ choices (*F*(6, 420) = 49.39, *p* < 0.001). For DS, responses overwhelmingly favored the character’s perspective (Option 1), which was selected significantly more often than the narrator’s perspective (Option 2; *β* = 12.25, *SE* = 0.93, *t* = 13.12, *p* < 0.001) or both perspectives (Option 3; *β* = 11.19, *SE* = 0.93, *t* = 11.99, *p* < 0.001). A comparable pattern was observed for FDS, where Option 1 was again chosen more frequently than Options 2–3 (all *p*s < 0.001). In contrast, for FIS, the narrator’s perspective (Option 2) was endorsed more often than the character’s perspective (Option 1; *β* = −6.22, *SE* = 0.93, *t* = −6.66, *p* < 0.001), and both perspectives (Option 3) were also preferred over the character’s perspective (*β* = −4.17, *SE* = 0.93, *t* = −4.46, *p* < 0.001). The difference between the narrator (Option 2) and both perspectives (Option 3) was not significant (*β* = 2.06, *SE* = 0.93, *t* = 2.20, *p* = 0.125).

### 2.3. Procedure

The experiment was conducted with an SR-Research (Ottawa, Canada) EyeLink 1000 desk-mounted eye tracker, recording monocularly (left eye) at 1000 Hz. Stimuli were presented using EyeTrack 0.7.1. Participants sat approximately 70 cm from the display (1280 × 960 resolution) in a dimly lit, sound-attenuated room. Text was presented in 32-point black font on a light gray background for ease of reading. A chinrest stabilized head position and maintained constant viewing distance. Participants were randomly assigned to one of six counterbalanced lists constructed using a Latin-square design. Each list contained experimental passages pseudorandomly intermixed with filler stories. The session began with task instructions, followed by standard nine-point calibration and validation. Each list consisted of 60 trials, divided into three blocks of 20. Short breaks were provided between blocks, and calibration/validation was repeated at the start of each block. Each trial comprised two parts (see Table 1): a context passage (Part A: providing background and character relationships) and a critical sentence (Part B: containing the target quotation). A drift-correction dot preceded each trial, located at the position of the first character of the context. Participants pressed the space bar while fixating the dot to trigger the display of the context, ensuring that reading always began at the first character. After reading the context, participants pressed the space bar to advance to the critical sentence. Following each passage, participants answered a yes/no comprehension question by pressing F for “Yes” or J for “No.” Questions were displayed centrally and ended with a question mark to signal response. 

### 2.4. Data Analysis

#### 2.4.1. Eye Movement

For the eye-movement analyses, data were extracted from the critical word (ROI 1) and the post-critical word region (ROI 2). We examined four standard measures. First Fixation Duration (FFD), defined as the duration of the very first fixation on a region, reflects the earliest stage of lexical access. Gaze Duration (GD), the sum of consecutive fixations from the first entry into a region until the eyes move out of it, is also sensitive to early lexical processing. Regression Path Duration (RPD), defined as the sum of fixations from the first entry into a region until the eyes exit it to the right (including any regressions to earlier text), indexes later integration and reanalysis processes. Finally, Total Reading Time (TRT), the sum of all fixations within a region across the entire trial, captures the overall processing load ([10]; [60]).

The raw data were extracted via EyeLink using DataViewer Software (version 4.1.477). Data cleaning followed established procedures ([10]; [41]; [49]). Trials with track loss (zero fixation time) were excluded. At the word level, fixations shorter than 80 ms or longer than 1200 ms were removed. For RPD, trials exceeding 5000 ms were also discarded ([60]). Trials with incorrect responses to comprehension questions were eliminated to ensure that participants had adequately processed the stimuli. Following data cleaning, linear mixed-effects models were fitted separately for FFD, GD, RPD, and TRT using the lme4 package in R (version 4.5.1; [8]). Fixed effects included *quotation type* (direct speech, free direct speech, free indirect speech) and *congruency* (congruent vs. incongruent). Maximal random-effects structures were specified initially ([4]), including by-*participant* and by-*item* random intercepts and slopes. If a model failed to converge, random slopes were removed stepwise until convergence was achieved. To further reduce noise, trials with reading times exceeding 2.5 SDs from the mean within each ROI were excluded, and new models were refitted. Additional outliers were identified through boxplot inspection of model residuals and removed. The final models were then refitted on the cleaned dataset, and results are reported based on these models.

#### 2.4.2. *Surprisal* and *Entropy* Estimation

Beyond the eye-movement measures, we incorporated computational estimates of *surprisal* and *entropy* to gain deeper insight into the mechanisms of predictive processing. *Surprisal*, defined as the negative log-probability of a word given its preceding context, indexes the degree of unexpectedness associated with linguistic input ([25]; [32]). *Entropy*, in contrast, captures the distributional uncertainty across all possible continuations, reflecting how strongly the context constrains upcoming words ([38]). Together, these measures provide a principled way to characterize the predictability and complexity of linguistic environments ([25]; [32]; [44]). The distinction between *surprisal* and *entropy* is important: *surprisal* quantifies the unexpectedness of the actually observed token, while *entropy* quantifies uncertainty over all possible next tokens. A manipulation can increase *entropy* by broadening the distribution of plausible continuations even when the eventual continuation is not extremely surprising.

In the present study, *surprisal* and *entropy* values were estimated for the critical word and the three subsequent positions using *GPT-2-Chinese-CLUEcorpus-small* ([66]). Although more advanced large language models are now available, GPT-2 was selected due to its open-source accessibility and its widespread application in psycholinguistic modeling, particularly for Chinese.

## 3. Results

### 3.1. Text Comprehension

Mean accuracy for comprehension question was 96%, indicating that participants had a clear understanding of the narrative texts.

### 3.2. Results of the Eye-Movement Analyses

#### 3.2.1. The Critical Region (ROI 1)

##### First Fixation Duration

There was a significant two-way interaction between quotation type and congruency (*χ*^2^ = 7.972, *p* = 0.019). Follow-up analysis to resolve the interaction showed that the incongruent word produced longer reading time than the congruent word (*β* = 0.067, *SE* = 0.030, *t* = 2.233, *p* = 0.026, Cohen’s *d* = 0.20) in DS, but no difference between the two conditions in either FDS or FIS, *p*s > 0.1 (See Figure 2A). In the other direction, FDS elicited more reading time than FIS (*β* = 0.079, *SE* = 0.031, *t* = 2.575, *p* = 0.027, Cohen’s *d* = 0.24) in congruent condition. No other comparison reached significant.

##### Gaze Duration

There was a significant main effect of quotation type (*χ*^2^ = 7.604, *p* = 0.022). Follow-up analyses revealed that DS yielded longer reading times than FIS (*β* = 0.073, *SE* = 0.028, *t* = 2.628, *p* = 0.024, Cohen’s *d* = 0.17). No other comparisons reached significance (*p*s > 0.10). No additional main effects or interactions were significant. See Figure 2B.

##### Total Reading Time

There was a significant main effect of congruency (*χ*^2^ = 30.593, *p* < 0.001), with incongruent sentences producing longer reading times than congruent sentences. A significant main effect of quotation type was also observed (*χ*^2^ = 26.311, *p* < 0.001). Follow-up analyses showed that DS elicited longer reading times than FIS (*β* = 0.174, *SE* = 0.034, *t* = 5.091, *p* < 0.001, Cohen’s *d* = 0.34) and that FDS also elicited longer reading times than FIS (*β* = 0.110, *SE* = 0.034, *t* = 3.227, *p* = 0.004, Cohen’s *d* = 0.21). However, there was no significant difference between DS and FDS (*β* = 0.064, *SE* = 0.034, *t* = 1.878, *p* = 0.145, Cohen’s *d* = 0.12). See Figure 2C.

##### Regression Path Duration 

There was a significant main effect of congruency (*χ*^2^ = 10.793, *p* = 0.001), with incongruent sentences yielding longer reading times than congruent sentences. A significant main effect of quotation type was also found (*χ*^2^ = 13.095, *p* = 0.001). Follow-up analyses indicated that both DS (*β* = 0.132, *SE* = 0.039, *t* = 3.393, *p* = 0.002, Cohen’s *d* = 0.22) and FDS (*β* = 0.109, *SE* = 0.039, *t* = 2.793, *p* = 0.015, Cohen’s *d* = 0.18) elicited longer reading times than FIS. No significant difference was observed between DS and FDS (*β* = 0.024, *SE* = 0.039, *t* = 0.605, *p* = 0.818, Cohen’s *d* = 0.04; See Figure 2D). A summary of eye-movement results is provided in Table 3. Detailed statistical outputs from the final linear mixed-model analyses of the eye-movement measures can be found in Appendix A. 

#### 3.2.2. The Post-Critical Region (ROI 2)

##### First Fixation Duration

No significant effects were observed.

##### Gaze Duration

No significant effects were observed.

##### Total Reading Time

There was a significant main effect of congruency (*χ*^2^ = 4.097, *p* = 0.043), with incongruent sentences producing longer reading times than congruent sentences. A significant main effect of quotation type was also observed (*χ*^2^ = 11.04, *p* = 0.004). Follow-up analyses revealed that DS yielded longer reading times than both FDS (*β* = 0.102, *SE* = 0.034, *t* = 2.983, *p* = 0.008, Cohen’s *d* = 0.20) and FIS (*β* = 0.095, *SE* = 0.034, *t* = 2.771, *p* = 0.016, Cohen’s *d* = 0.19). However, there was no significant difference between FDS and FIS (*β* = 0.007, *SE* = 0.034, *t* = 0.218, *p* = 0.974, Cohen’s *d* = 0.01). See Figure 3A.

##### Regression Path Duration

There was a significant main effect of congruency (*χ*^2^ = 7.741, *p* = 0.005), with incongruent sentences eliciting longer reading times than congruent sentences. A significant main effect of quotation type was also observed (*χ*^2^ = 13.439, *p* = 0.001). Follow-up analyses showed that both DS (*β* = 0.139, *SE* = 0.040, *t* = 3.430, *p* = 0.002, Cohen’s *d* = 0.23) and FDS (*β* = 0.115, *SE* = 0.040, *t* = 2.859, *p* = 0.012, Cohen’s *d* = 0.19) produced longer reading times than FIS. However, no significant difference was found between DS and FDS (*β* = 0.024, *SE* = 0.040, *t* = 0.585, *p* = 0.828, Cohen’s *d* = 0.04). See Figure 3B.

### 3.3. LLM Results

#### 3.3.1. *Surprisal*

For the critical word, there was a significant main effect of congruency (*χ*^2^ = 62.671, *p* < 0.001), with the incongruent condition eliciting greater *surprisal* than the congruent condition (Figure 4A). Neither the main effect of quotation type (*χ*^2^ = 0.015, *p* = 0.993) nor the interaction between consistency and quotation type (*χ*^2^ = 0.026, *p* = 0.987) reached significance. For the post-1 word, there was a significant main effect of quotation type (*χ*^2^ = 62.61, *p* < 0.001). Post hoc analyses revealed that DS produced greater *surprisal* than FDS (*β* = 0.475, *SE* = 0.058, *t* = 8.172, *p* < 0.001, Cohen’s *d* = 1.20) and FIS (*β* = 0.337, *SE* = 0.058, *t* = 5.778, *p* < 0.001, Cohen’s *d* = 0.85). In addition, FIS yielded larger *surprisal* than FDS (*β* = 0.138, *SE* = 0.058, *t* = 2.367, *p* = 0.049, Cohen’s *d* = 0.35). For the post-2 word, quotation type exerted a significant main effect (*χ*^2^ = 135.16, *p* < 0.001). DS elicited greater *surprisal* than both FDS (*β* = 0.990, *SE* = 0.077, *t* = 12.923, *p* < 0.001, Cohen’s *d* = 1.89) and FIS (*β* = 0.738, *SE* = 0.077, *t* = 9.587, *p* < 0.001, Cohen’s *d* = 1.41). FIS also produced higher *surprisal* than FDS (*β* = 0.252, *SE* = 0.077, *t* = 3.268, *p* = 0.004, Cohen’s *d* = 0.48). For the post-3 word, a significant main effect of quotation type was observed (*χ*^2^ = 49.199, *p* < 0.001). DS yielded higher *surprisal* than both FDS (*β* = 0.342, *SE* = 0.049, *t* = 7.055, *p* < 0.001, Cohen’s *d* = 1.03) and FIS (*β* = 0.257, *SE* = 0.049, *t* = 5.293, *p* < 0.001, Cohen’s *d* = 0.77). However, *surprisal* did not differ between FDS and FIS (*β* = 0.085, *SE* = 0.049, *t* = 1.762, *p* = 0.185, Cohen’s *d* = 0.25).

#### 3.3.2. *Entropy*

For the critical word, there was a significant main effect of congruency (*χ*^2^ = 26.615, *p* < 0.001), with incongruent items producing greater *entropy* than congruent items (Figure 4B). Neither the main effect of quotation type (*χ*^2^ = 0.022, *p* = 0.989) nor the interaction (*χ*^2^ = 0.020, *p* = 0.989) was significant. For the post-1 word, no effects were significant. For the post-2 word, there was a significant main effect of quotation type (*χ*^2^ = 7.305, *p* = 0.026). Post hoc tests showed that DS elicited greater *surprisal* than FIS (*β* = 0.074, *SE* = 0.030, *t* = 2.464, *p* = 0.038, Cohen’s *d* = 0.36), while other contrasts were non-significant. For the post-3 word, no significant effects emerged. A summary of *surprisal* and *entropy* results is provided in Table 4. Detailed statistical outputs from the final linear mixed-model analyses of the LLM metrics can be found in Appendix A.

## 4. Discussion

This study examined how Chinese readers process direct speech (DS), free direct speech (FDS), and free indirect speech (FIS) during natural reading of classical Chinese novels, combining fine-grained eye-tracking measures with *surprisal* and *entropy* estimates from a GPT-2 model. Behavioral results showed that DS reliably elicited longer fixation times than FIS (with FDS patterning closer to DS in most eye-movement indices), while perspective incongruency (incongruent address vs. congruent address) produced consistent slowing, especially in later processing measures. The computational analyses revealed a related but not identical picture: *surprisal* showed robust effects (DS > FDS/FIS) across post-critical positions, whereas *entropy* effects were concentrated at the critical word, where incongruency produced higher *entropy* but otherwise yielded few significant contrasts. Below we synthesize these outcomes, discuss why *surprisal* and *entropy* can produce different patterns, and consider theoretical implications for narrative perspective-taking and for comparisons between humans and current LLMs within the specific context of Chinese classical literature.

### 4.1. Direct Speech, Perspective Shifts, and Processing Cost

Across eye-movement measures, DS imposed greater processing cost than FIS, consistent with the pattern reported in prior work: while DS increases immediacy and perceptual simulation, it also requires additional integration and perspective-taking effort. From an embodied cognition perspective ([6]), this increased cost may reflect the cognitive resources required to maintain rich sensorimotor simulations of the character’s voice, gesture, and emotional state. The activation of these perceptual symbols systems creates a more vivid but also more demanding representational format compared to the more abstract processing associated with FIS.

*Surprisal* estimates from GPT-2 similarly assigned higher unexpectedness to DS continuations, suggesting that, at the probabilistic level captured by the model, direct quotation produces less predictable continuations. The heightened *surprisal* in DS suggests that LLMs, like human readers, treat DS as less predictable and more information-rich. From a computational perspective, DS involves a shift in perspective-taking that increases contextual variability, making subsequent words harder to predict ([30]; [34]). This aligns with Theory of Mind accounts, which link DS to additional inferencing demands as readers must construct and maintain representations of the speaker’s mental states and intentions ([2]). Importantly, our study extends such findings from sentence-level experiments to narrative discourse, showing that direct quotation in classical narrative also triggers perspective shifts and processing costs.

Notably, FDS patterned closely with DS in human eye movements in both early and late measure (except total reading time at the post-critical word position) but diverged from DS in LLM *surprisal* metrics. This suggests that for readers, the omission of quotation markers does not substantially reduce processing costs, since both DS and FDS require a perspective shift from narrator to character. The embodied simulation processes activated by both forms may be similar, as readers must still construct a representation of the character’s voice and perspective regardless of punctuation markers. By contrast, LLMs appear more dependent on surface cues such as quotation marks and less sensitive to perspectival alignment. Thus, while humans flexibly accommodate FDS, LLMs show divergence in how they process DS relative to FDS and FIS.

### 4.2. Free Indirect Speech: Absence of a Dual-Voice Effect

The findings for FIS present a more nuanced picture. In early measures (FFD), FIS showed no congruency effect, suggesting that readers do not immediately resolve perspectival ambiguity at the lexical level. In later measures (i.e., regression path duration, total reading time), FIS patterned more like DS and FDS, indicating that readers allocate additional resources when integrating FIS into the broader discourse representation.

Crucially, we found no robust evidence for an interaction between quotation type and congruency across most measures in our specific manipulation using *address terms* versus *proper names* in classical Chinese texts. Instead, a main effect of congruency emerged, suggesting that, at least for this particular operationalization in Chinese classical novels, FIS does not trigger the distinctive “dual-voice” processing predicted by the dual-voice account ([9]; [46]). Rather, our findings align more closely with single-voice accounts, which hold that FIS is primarily processed from the narrator’s stance while still conveying character-specific flavor through expressive markers ([3]; [45]).

These findings also highlight a divergence between offline interpretive studies, which frequently report mixed or dual-voice judgments ([9]; [51]), and online processing evidence, which shows little trace of simultaneous dual voice in our specific experimental context. A likely explanation is that dual-voice interpretations emerge retrospectively during reflective reading, when readers have sufficient time to deliberate, but not during immediate processing, where cognitive resources are limited. Alternatively, what appears as FIS in classical Chinese texts might be better characterized as protagonist projection or viewpoint shift ([1]), phenomena that create perspective effects without true dual voicing. Abrusán’s framework distinguishes viewpoint shifts—where a narrator adopts the perceptual or epistemic perspective of a character without representing their speech—from genuine FIS, which involves dual voicing. This distinction may be particularly relevant for Chinese classical literature, where narrative perspective is often signaled through subtle shifts in evaluative language, spatial deixis, and the use of evidential markers (e.g., 只见 ‘only see’, 但见 ‘but see’) that anchor the reader’s viewpoint to a character’s perceptual field without explicit quotation frames. Unlike in Western literature, where FIS is typically marked by tense backshifting and pronominal shifts, Chinese lacks grammatical tense and relies heavily on contextual and lexical cues for perspective signaling. These language-specific features may explain why FIS in classical Chinese novels patterns more like single-voice narration with embedded viewpoint shifts rather than true dual voicing. This possibility deserves further investigation with materials specifically designed to distinguish these phenomena.

### 4.3. Implications for Embodied Cognition and Theory of Mind

Our findings contribute to understanding the role of embodied cognition and Theory of Mind in narrative processing. The increased processing costs for DS align with embodied accounts suggesting that vivid perceptual simulation, while enhancing engagement, demands cognitive resources ([7]). The pattern whereby DS showed both higher *surprisal* and longer reading times suggests that embodied simulation and predictive processing may be linked: the richer the simulation, the more variable the possible continuations, leading to both processing cost and predictive uncertainty.

The absence of clear dual-voice effects for FIS in our study raises questions about the relationship between linguistic form and Theory of Mind activation. If FIS truly required simultaneous tracking of two mental perspectives, we might expect distinctive processing patterns or interactions with our congruency manipulation. The fact that FIS patterned similarly to other quotation types in showing main effects of congruency suggests that, at least in online processing, readers may default to a single predominant perspective even when the linguistic form theoretically allows for dual voicing.

This has implications for understanding how Theory of Mind operates in narrative comprehension. Rather than being automatically triggered by specific linguistic forms, perspective-taking may be a more flexible process influenced by factors including genre conventions, cultural expectations, reader expertise, and the specific nature of the perspective cues available in the text ([29]).

### 4.4. Comparing Human and LLM Processing

By combining eye-tracking with LLM metrics, this study contributes to the growing dialog between human cognition and artificial language processing. The parallels—such as greater costs for DS relative to FIS and for incongruent over congruent reference forms—show that LLMs capture certain probabilistic aspects of discourse processing. Yet the divergences, particularly in FDS, suggest that LLMs remain largely surface-driven and limited in perspective-taking. The FIS condition is especially revealing: although processing FIS is cognitively less demanding than DS, readers do not seem to maintain two perspectives simultaneously in our experimental context. LLMs similarly show no evidence of dual-voice sensitivity, though likely for different reasons—lacking the Theory of Mind capacities that would enable true perspective representation.

Taken together, these findings position quotation processing as a novel test case for assessing the cognitive plausibility of LLMs. While models approximate predictive uncertainty at the lexical level, they fall short in simulating deeper coherence-driven expectations and the embodied simulation processes that characterize human narrative understanding ([38]). This underscores that human narrative understanding is not merely a matter of statistical prediction but involves perspective simulation grounded in embodied experience and context integration beyond current LLM capacities.

### 4.5. Why Surprisal and Entropy Gave Different Empirical Pictures

Importantly, *surprisal* and *entropy* did not mirror each other in our LLM analyses. *Surprisal* produced clearer and more widespread differences between quotation types across the post-critical window, whereas *entropy’s* effects were largely restricted to the critical word (where congruency reliably increased *entropy*). This divergence is informative rather than problematic, because the two measures index distinct computational properties: *surprisal* quantifies the unexpectedness of the actually observed token, whereas *entropy* quantifies the model’s uncertainty over all possible next tokens (the distributional spread of alternatives; [25]; [32]; [38]). Thus, a manipulation can increase *entropy*—by broadening the distribution of plausible continuations at a particular point—even when the eventual observed continuation is not extremely surprising (low *surprisal*), and conversely an unlikely observed token can produce high *surprisal* even if the overall distribution was already narrow. Concretely in our data, perspective incongruency appears to widen the model’s distribution of possible continuations at the critical word (raising *entropy*), but only later (in the post-critical context) do different continuations actually materialize with systematically differing probabilities (hence the stronger *surprisal* contrasts downstream). Put differently, incongruency initially increases uncertainty about what should follow; as further context resolves that uncertainty, the model’s *surprisal* profile diverges by quotation type. Additionally, the model’s training distribution and sensitivity to surface markers (e.g., quotation symbols, punctuation, or explicit reporting verbs) also produce position-dependent effects: when an explicit cue appears earlier, post-critical *surprisal* differences grow, whereas *entropy* at the critical word may better reflect immediate ambiguity induced by incongruent referential forms.

The complementarity of *surprisal* and *entropy* provides a richer picture than either metric alone. Human readers appear to react to both uncertainty and realized unpredictability, but not in exactly the same way as a text-only LLM: *surprisal* aligns with realized lexical unpredictability, whereas *entropy* indexes momentary distributional uncertainty that may drive early monitoring or reanalysis processes. The pattern that *entropy* effects in the model were strongest at the critical word while *surprisal* effects grew later suggests that human readers may similarly experience a transient increase in expectation uncertainty at the referent, which then leads to increased integration costs when subsequent words disfavor the initial expectations. However, because current LLMs lack grounded pragmatic and social knowledge, they do not fully model perspective taking (especially for FDS and FIS), which contributes to divergences between probabilistic signals and human behavior.

### 4.6. Limitations

Despite these insights, several limitations must be acknowledged. First and most importantly, our materials were drawn from classical Chinese novels, which, while ecologically valid for studying an important literary tradition, may not generalize to contemporary texts or other genres. The specific narrative conventions, stylistic features, and cultural associations of these classical texts may influence processing in ways that differ from modern fiction or other discourse types.

Second, our operationalization of congruency relied solely on *address terms* versus *proper names*, which represents only one dimension of perspectival marking. This manipulation, while culturally salient in Chinese, may not capture the full complexity of perspective cues available in narrative texts. Other manipulations—such as dialectal features, emotional expressions, or deictic markers—might yield different results. The absence of dual-voice effects in our study does not rule out their existence under different conditions. Cultural factors specific to Chinese kinship terminology and address forms may play a role. The strong pragmatic violations associated with using proper names for family members in Chinese may override subtler perspective effects that might emerge in other linguistic contexts. Third, although all participants reported familiarity with the Four Great Classical Novels through formal education, individual reading histories were not systematically assessed or controlled. Given the canonical status of these works in Chinese education and culture, variation in familiarity or emotional engagement with particular characters and episodes may have influenced reading patterns in ways not captured by our design. Fourth, while LLM metrics provided a useful computational benchmark, current models are trained on text corpora without perceptual or social grounding, limiting their comparability to human cognition. Finally, our sample consisted entirely of university students, who may possess reading strategies and literary expertise not representative of the broader population. Importantly, we did not assess individual differences in Theory of Mind (ToM) ability, working memory (WM) capacity, or reading expertise—factors that have been shown to modulate text comprehension ([28]; [37]). Such individual differences could plausibly influence quotation processing, particularly given that perspective-taking and narrative comprehension rely on ToM mechanisms. The absence of these measures in the present study therefore limits our ability to determine whether the observed patterns generalize across readers or whether specific subgroups (e.g., those with higher ToM ability or greater literary expertise) might exhibit distinct processing profiles. Future research should systematically investigate how these individual factors interact with quotation type and perspective congruency, especially in contexts where more nuanced perspective manipulations may reveal reader-specific sensitivity.

## 5. Conclusions

By examining quotation processing in classical Chinese novels, this study extends narrative research and evaluates theoretical claims about perspective-taking within a specific cultural and literary context. Combining eye-tracking with LLM-based *surprisal* and *entropy* measures, the findings show that while LLMs capture certain predictive patterns, they diverge from humans in handling perspectival shifts, particularly in FIS. Our results provide evidence that in the specific context of classical Chinese novels with *address term*/*proper name* manipulations, FIS does not show the clear dual-voice effects predicted by some theoretical accounts. However, this finding should be interpreted cautiously given the specific nature of our materials and manipulation. This dual approach demonstrates that human readers process quotations through mechanisms involving embodied simulation and Theory of Mind processes not fully replicated in current models, underscoring both the complexity of narrative comprehension and the limitations of LLMs in modeling human discourse understanding. The study highlights the importance of considering cultural context, genre conventions, and specific operational definitions when investigating narrative perspective, and suggests that universal claims about quotation processing should be tempered by attention to language-specific and context-specific factors.

## Figures and Tables

**Figure 1 behavsci-15-01650-f001:**
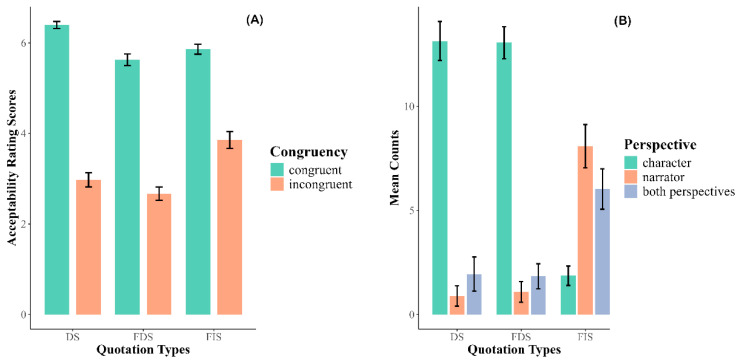
Descriptive statistics of two pre-tests. (**A**) The mean rates of acceptability judgment; (**B**) The counts of responses for the narrator’s perspective, the character’s perspective, and both perspectives. DS: direct speech; FDS: free direct speech; FIS: free indirect speech.

**Figure 2 behavsci-15-01650-f002:**
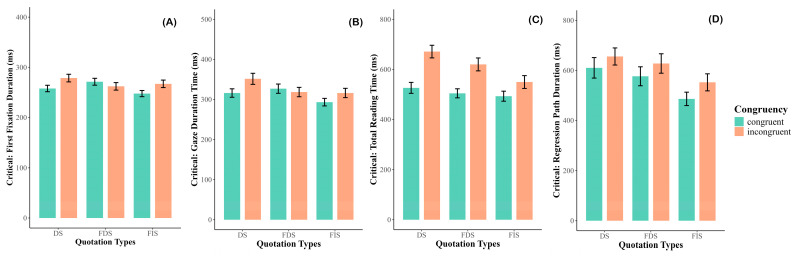
Results of the critical region. (**A**) First Fixation Duration; (**B**) Gaze Duration; (**C**) Total Reading Time; (**D**) Regression Path Duration. DS: direct speech; FDS: free direct speech; FIS: free indirect speech.

**Figure 3 behavsci-15-01650-f003:**
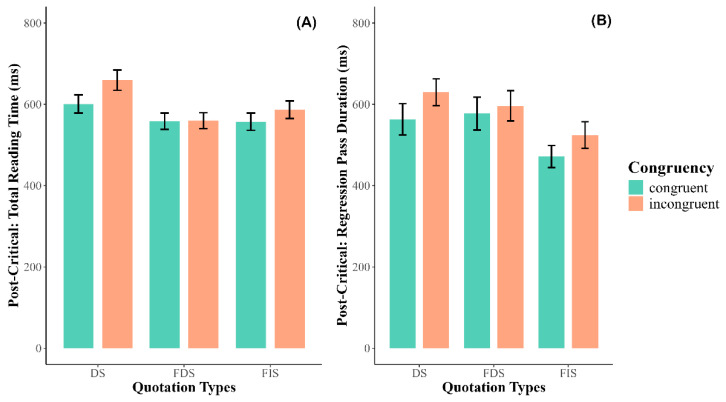
Results of the post-critical region. (**A**) Total Reading Time; (**B**) Regression Path Duration. DS: direct speech; FDS: free direct speech; FIS: free indirect speech.

**Figure 4 behavsci-15-01650-f004:**
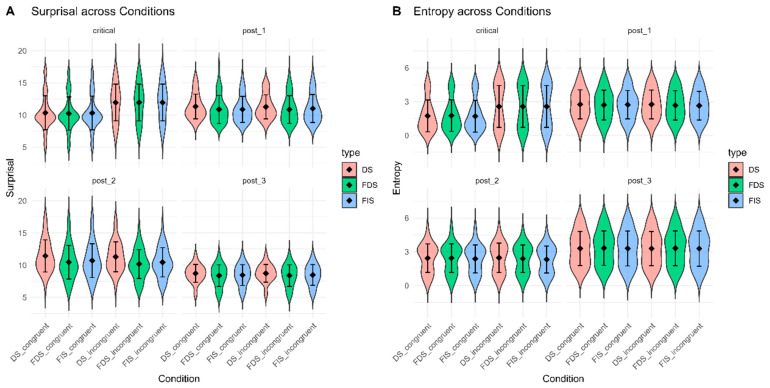
*Surprisal* and *Entropy* values. (**A**) *Surprisal* values for the critical word and the three subsequent word positions. (**B**) *Entropy* values for the critical word and the three subsequent word positions.

**Table 1 behavsci-15-01650-t001:** Sample stimuli in each of the experimental conditions.

	Examples
**Part A:** Context	哪吒率先施法，变出三头六臂，朝那妖怪打去，没想到那妖怪只取出一个圈，往空抛起，就将哪吒的兵器尽数套来。哪吒之父李靖见状，便让哪吒在身后等待，拿起宝塔便要施法，谁知被那妖怪把宝塔也给收了去，哪吒见他父王居然也败了，觉得非常不可思议。 Nezha took the lead in casting magic, with his three heads and six arms appearing to attack the monster. Unexpectedly, the monster only took out a circle and threw it into the air. Then Nezha’s weapon was taken by the monster. Nezha’s father, LiJing asked Nezha to wait behind him. He picked up the pagoda and tried to cast magic, but his pagoda was also taken by the monster soon. Nezha felt incredible that his father had also been defeated.
**Part B:** Critical sentence	/ROI 1/ROI 2/
DS-congruent	哪吒大吃一惊，他说：“我真是没有想到，竟连/***父王*/也吃**/败仗/了。/” Nezha was shocked. “I can’t believe it, ***father* has also** been defeated,” he said.
DS-incongruent	哪吒大吃一惊，他说：“我真是没有想到，竟连/***李靖*/也吃**/败仗/了。/” Nezha was shocked. “I can’t believe it, ***Lijing* has also** been defeated,” he said.
FDS-congruent	哪吒大吃一惊，我真是没有想到，竟连/***父王*/也吃**/败仗/了。/ Nezha was shocked. I can’t believe it, ***father* has also** been defeated.
FDS-incongruent	哪吒大吃一惊，我真是没有想到，竟连/***李靖*/也吃**/败仗/了。/ Nezha was shocked. I can’t believe it, ***Lijing* has also** been defeated..
FIS-congruent	哪吒大吃一惊，他真是没有想到，竟连/***父王*/也吃**/败仗/了。/ Nezha was shocked. He couldn’t believe it, ***father* had also** been defeated.
FIS-incongruent	哪吒大吃一惊，他真是没有想到，竟连/***李靖*/也吃**/败仗/了。/ Nezha was shocked. He couldn’t believe it, ***Lijing* had also** been defeated.

Note: ROI 1 corresponds to the bolded critical word, and ROI 2 to the post-critical word.

**Table 2 behavsci-15-01650-t002:** Counts of responses for the narrator’s perspective, the character’s perspective, and both perspectives.

	Character’s Perspective	Narrator’s Perspective	Both Perspective	Total
DS	473	32	70	575
FDS	470	39	66	575
FIS	67	291	217	575

**Table 3 behavsci-15-01650-t003:** Results of linear mixed-model analyses of eye-movement measures.

Eye- Movement	ROI	Congruency	Quotation Type	DS > FDS	DS > FIS	FDS > FIS
FFD	ROI 1 (critical)	Yes (DS)	n.s.	n.s.	n.s.	Yes
GD	ROI 1 (critical)	n.s.	Yes	n.s.	Yes	n.s.
TRT	ROI 1 (critical)	Yes	Yes	n.s.	Yes	Yes
RPD	ROI 1 (critical)	Yes	Yes	n.s.	Yes	Yes
FFD	ROI 2 (post-critical)	n.s.	n.s.	-	-	-
GD	ROI 2 (post-critical)	n.s.	n.s.	-	-	-
TRT	ROI 2 (post-critical)	Yes	Yes	Yes	Yes	n.s.
RPD	ROI 2 (post-critical)	Yes	Yes	n.s.	Yes	Yes

Note: FFD: First Fixation Duration; GD: Gaze Duration; TRT: Total Reading Time; RPD: Regression Path Duration; ROI: Region of Interest.

**Table 4 behavsci-15-01650-t004:** Results of linear mixed-model analyses of *surprisal* and *entropy* derived from LLM metrics.

LLM Metrics	Word Position	Consistency	Quotation Type	DS > FDS	DS > FIS	FIS > FDS
*Surprisal*	Critical	Yes	n.s.	–	–	–
	Post-1	n.s.	Yes	Yes	Yes	Yes
	Post-2	n.s.	Yes	Yes	Yes	Yes
	Post-3	n.s.	Yes	Yes	Yes	n.s.
*Entropy*	Critical	Yes	n.s.	–	–	–
	Post-1	n.s.	n.s.	–	–	–
	Post-2	n.s.	Yes	n.s.	DS < FIS	n.s.
	Post-3	n.s.	n.s.	–		–

Note: “Post-1” refers to the first word following the critical word; the other two labels follow the same pattern.

## Data Availability

The Experimental materials, data files, and analysis scripts can be found at the project’s OSF page. https://osf.io/4uthb/ (accessed on 19 September 2025).

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
