# Peer review of "How Quotation Types Shape Classic Novel Reading in Chinese: A Comparison Between Human Eye-Movements and Large Language Models"

_behavsci, 2025, doi:10.3390/bs15121650_

Round 1

Reviewer 1 Report

Comments and Suggestions for Authors

This is a very interesting and well-designed study. A major strength of this study is its combination of two methods: using eye-tracking to observe people’s real-time reading processes, and using a LLM to calculate the “surprisal” and “entropy” of the text. The research method is solid, the results are clear, and I do think it makes a valuable contribution to the current topic. However, there are still some major points that should be improved, and some open questions that need to be answered.

Introduction:

  1. Why not include free indirect speech (FIS) as one of the keywords? It is a central concept in the present study.

  1. Using address terms (e.g., “father”) vs. proper names (e.g., “Lijing”) to manipulate perspective congruency is a clever design. However, the rationale for treating address terms as markers of the character’s perspective even in FIS could be explained more, especially since the pre-test showed mixed participant judgments on this.

  1. The study requires further theoretical foundations in both Introduction and Discussion. “Embodied cognition framework”(Barsalau, 1999, 2008) and “Theory of Mind” are classic models that one cannot avoid when talking about DS and IS, but the authors did not mention them.

Methodology:

  1. For the eye-tracking experiment, how did the authors choose the areas of interest (AOI)? In the text, these are sometimes referred to as “critical sentences”, “critical words”, “post-words”, “post-critical words” or just “regions”. Even after thoroughly reading the paper, it was still unclear which AOI had the eye-movement data registered, nor if they were congruent words, pronouns, or quotation marks. If the study used the software EyeDoctor like in previous studies (Yao & Scheeper, 2011), it needs to be described and explained.

  1. The method used to control the LLM unbiased baseline was very appropriate. However, the distinction between Surprisal and Entropy must be further explained in the Methodology section.

  1. Since the authors are measuring readers impression of reading characters speech in a narrative, the relation between participants and material must be taken into consideration. The chosen texts were described as classical narrative, but they have been adapted to modern literature. In what sense, then, are they different from normal narrative? Additionally, since these classical texts are part of Chinese education, participants likely have background knowledge of the literature and an emotional engagement to some of the characters that can explain some of the results like in previous studies (Dixon & Bortolussi, 1996). Then, why has the methodology excluded the evaluation of participants emotional engagement with the protagonists? The reason needs to be explained.

Results:

In Sections 3.2 and 3.3, the statistical symbols in the text are not italicized. For the eye-tracking results, it is better to summarize the mixed-effects model analyses in tables as in Table 3, with the relevant statistics such as β, SE, t/z, p, and χ² values.

Discussion:

  1. The discussion could more directly address why the findings of this study (no strong dual-voice effect in real-time reading) differ from some prior offline questionnaire studies (which found that people retrospectively report a dual-voice effect. e.g., Bray, 2007; Sotirova, 2006). A clearer explanation of this discrepancy would be more helpful for theoretical development.

  1. In section 4.1 the sentence “…DS imposed greater processing cost than FIS, consistent with the pattern reported in prior work: while DS increases immediacy and perceptual simulation, it also requires additional integration and perspective-taking effort (Köder et al., 2015; Yao & Scheepers, 2011; Li et al., 2024).” seems inaccurate. Neither the study by Köder et al. (2015) nor the study by Yao and Scheepers (2011) researched FIS.

  1. In section 4.1 the sentence “…This aligns with cognitive cost accounts, which link DS to additional inferencing demands (Köder et al., 2015; Yao & Scheepers, 2011).” seems inaccurate. On the one hand, Yao and Scheepers (2011) results showed just the contrary, that DS is linked to fewer inferencing demands. On the other hand, Köder et al. (2015) results on DS linked to more inferencing demands are related with their experimental design, which was focused on the pronouns and contradict previous consideration of DS being more basic, as it is first acquired by children. These two references need to be corrected and clarified.

  1. In section 4.2, the results demonstrating that LLMs process FIS differently from humans can be interpreted through a similar lens as the findings of Oh and Schuler (2022). Specifically, the LLM struggles to interpret unmarked speech because they operate without a mental model of reality that humans use to resolve ambiguity.

  1. In section 4.3, the results from the LLMs are consistent with those of Oh and Schuler (2022), who also found that the predictors of “Surprisal” and “Entropy” exhibited different patterns across datasets.

  1. The use of classic novels is excellent, but a brief discussion on whether these findings might generalize to modern fiction or other genres of writing would be beneficial.

References:

The following reference is wrong or does not exist:

Yao, B., & Scheepers, C. (2011). Contextual modulation of reading direct versus indirect speech: Evidence from eye movements. 717 Language and Cognitive Processes, 26(3), 402–422. ???

Reviewer 2 Report

Comments and Suggestions for Authors

This study makes several valuable contributions to our understanding of quotation processing in narrative comprehension. The use of eye-tracking methodology to examine three types of quotation in Chinese represents important empirical work in an understudied language. The integration of computational metrics alongside human behavioral data is innovative and provides complementary perspectives on processing mechanisms. The materials drawn from classical Chinese novels offer ecological validity, and the within-subjects experimental design with adequate statistical power demonstrates methodological rigor. The finding that free indirect speech doesn't show special dual-voice processing in Chinese adds meaningful data to ongoing theoretical debates.

However, the most critical issue requiring attention is the mismatch between the scope of your evidence and the breadth of your theoretical claims. The study tests one specific manipulation using address terms versus proper names in classical Chinese texts, yet the conclusions make broad claims about FIS processing universally. This needs substantial revision. The discussion and conclusions should acknowledge that findings apply specifically to this context rather than to FIS processing in general. Consider reframing your contribution as evidence against dual-voice processing of FIS in classical Chinese novels specifically, which would make the claims more defensible while preserving the value of your findings.

The literature review and theoretical framework require significant expansion to properly situate this work within existing scholarship. The current reference list relies heavily on sources from the 1980s and 1990s for FIS theory, missing important developments from the past decade. Given that you're studying Chinese texts, the limited engagement with Chinese linguistic scholarship is particularly problematic. The computational component would benefit from deeper grounding in computational narratology literature beyond the basic references to surprisal and entropy. Additionally, since you're using the Four Great Classical Novels as materials, some engagement with literary criticism on narrative techniques in these works would strengthen the theoretical foundation.

Moreover, the presentation of results needs reorganization for clarity. A summary table showing all effects across the different eye-tracking measures would help readers grasp the overall pattern of findings. Effect sizes should be reported alongside p-values to indicate practical significance.  

Several methodological decisions require better justification. Why was the address terms versus proper names manipulation chosen over other possible ways to test perspective congruency? What are the limitations of this specific operationalization? Why use GPT-2 when newer models are available? These choices may be reasonable, but readers need to understand the rationale. Additionally, the extent to which classical Chinese novels represent typical narrative processing should be discussed, as this has implications for generalizability.

The discussion section needs substantial refinement to provide a more balanced interpretation of findings. Alternative explanations for the absence of dual-voice effects should be thoroughly explored. Perhaps the specific manipulation used, the nature of classical Chinese texts, or cultural reading conventions could explain the results rather than FIS lacking dual-voice properties generally. The findings need to be reconciled with previous studies showing FIS effects in other languages and contexts. The theoretical implications should be stated more cautiously, acknowledging the specific constraints of what was tested. Adding a section on practical applications would help readers understand the broader significance of this work.

Comments on the Quality of English Language

The manuscript would benefit from comprehensive language editing to improve clarity and readability. While the research content is generally comprehensible, several issues with expression, grammar, and style interfere with effective communication of the findings.

Round 2

Reviewer 1 Report

Comments and Suggestions for Authors

I have some minor points.

Line 105. In APA style, multiple citations by different authors within the same set of parentheses should be listed in alphabetical order by the first author’s last name. If you are using this criterion, it is necessary to review the references within brackets throughout the paper.

Line 272 and 284. The methodology section is sufficient to mention the participants' familiarity with the four classic texts. It is not necessary to mention limitations during the methodology description; they can be mentioned later in the limitations section.

Table 1. Part B. The “ROI 1” has been described as the critical word in bold. However, personal pronouns such as “I” and “He” in the Chinese version of the stimuli have also been bolded. This can lead to confusion, as it seems as though there are two “ROI 1”.

In addition, “ROI 2”, expressed as the post-critical word region, does not specify whether it is composed of two or more words. This must be taken into consideration, bearing in mind that in some of the examples, “ROI 1” is followed by one-word characters.

Table 3. The table lacks statistical values or significance levels, and if necessary, it could be divided into several separate tables. Using only “Yes” or “n.s.” to indicate the results is concise but lacks transparency. As I mentioned in my previous comments, it is recommended to report statistical values (e.g., t, β, SE, p). At present, these statistics appear only partially in the text, but tables should be relatively self-contained, allowing readers to understand the main results directly from them.
In addition, “(DS: Incon > Con)” appears inconsistent with the other rows and does not indicate whether the effect is significant.

It would also be clearer to display “ROI 1” and “ROI 2” as separate groups, or to use a small blank line or horizontal rule to distinguish between the two ROI sections.

Line 597. Although this reference is a strong point in your discussion, I found no mention of “direct speech” in the work of Mason and Just (2006). The sentence could be rephrased to refer to the process of “mentalising” another person or a third person while reading a text containing direct speech.

Line 633. The explanation of the results based on the use of “viewpoint shifts” in Chinese classical literature is compelling. However, I found no mention of language-specific variations in the use of this literary device, particularly in Chinese, in Abrusán’s (2021) description. The sentence could be completed by describing the specific characteristics of Chinese literature and their possible relation to “viewpoint shifts”, as described by this author. In addition, if the use of FIS in Chinese classical literature is particularly distinctive, this could be further mentioned in more detail in the introduction.

Reviewer 2 Report

Comments and Suggestions for Authors

The authors have successfully enhanced the theoretical framework by adding explicit discussion of embodied cognition (Barsalou, 1999, 2008) throughout the introduction and discussion. They've also integrated the Theory of Mind framework (Kim, 2016, 2020; Mason & Just, 2006) and connected these frameworks to explain why direct speech incurs higher processing costs. This addresses one of the main concerns about theoretical grounding.

Regarding methodological clarifications, the authors now clearly explain the address terms versus proper names manipulation with detailed examples in Table 1 and comprehensive explanation on pages 6-7. They've added justification for why this manipulation tests perspective congruency in the Chinese cultural context and acknowledged that this represents "one specific way to test perspective effects," showing appropriate methodological transparency.

The justification for materials selection has been strengthened. The authors explain why the Four Great Classical Novels were chosen as canonical works with rich quotation patterns, while acknowledging limitations that participants' background knowledge and emotional engagement with specific characters may have influenced processing. This shows a more nuanced understanding of their methodological choices.

The requested power analysis has been added using G*Power on page 6, justifying the sample size of 37 participants (33 after exclusions). The authors also now explain clearly why GPT-2 was chosen for the computational modeling, citing its open-source availability, computational tractability, and widespread use in psycholinguistic research. They've added section 4.5 explaining why surprisal and entropy gave different empirical pictures, which helps readers understand the complementary nature of these measures.

Statistical reporting has been improved with Cohen's d effect sizes added throughout the results section and a comprehensive Table 3 summarizing all eye-movement results. The literature review now includes recent references from 2021-2025 and better integrates computational narratology literature while connecting to broader theoretical debates.

The cultural context is now well addressed with discussion of Chinese-specific features such as the absence of tense marking and frequent subject omission. The authors explain the cultural significance of kinship terms versus proper names in Chinese, which helps justify their experimental manipulation.

A comprehensive limitations section (4.6, pages 17-18) has been added that acknowledges ecological validity concerns and discusses generalizability issues honestly. This demonstrates scholarly rigor and appropriate interpretation of results.

However, some areas could still be improved. While individual differences are mentioned in the limitations, there's still no actual analysis of Theory of Mind abilities, working memory, or reading expertise. Participants' familiarity with the Four Great Classical Novels was acknowledged but not systematically assessed, which the authors note as a limitation. There's also no comparison with modern texts, though this is acknowledged as a limitation for future work.

Comments on the Quality of English Language

The manuscript would benefit from comprehensive language editing to improve clarity and readability. While the research content is generally comprehensible, several issues with expression, grammar, and style interfere with effective communication of the findings.
